# TRAINING BY VANILLA SGD WITH LARGER LEARNING RATES

## ABSTRACT

The stochastic gradient descent (SGD) method, first proposed in 1950's, has been the foundation for deep-neural-network (DNN) training with numerous enhancements including adding a momentum or adaptively selecting learning rates, or using both strategies and more. A common view for SGD is that the learning rate should be eventually made small in order to reach sufficiently good approximate solutions. Another widely held view is that the vanilla SGD is out of fashion in comparison to many of its modern variations. In this work, we provide a contrarian claim that, when training over-parameterized DNNs, the vanilla SGD can still compete well with, and oftentimes outperform, its more recent variations by simply using learning rates significantly larger than commonly used values. We establish theoretical results to explain this local convergence behavior of SGD on nonconvex functions, and also present computational evidence, across multiple tasks including image classification, speech recognition and natural language processing, to support the practice of using larger learning rates.

## 1 INTRODUCTION

We are interested in minimizing a function $f : \mathbb{R}^d \to \mathbb{R}$ with an expectation form:

$$\min_{x \in \mathbb{R}^d} f(x) := \mathbb{E}_\xi[K(x, \xi)], \tag{1a}$$

where the subscript indicates expectation is computed on the random variable $\xi$. Specially, if the $\xi$ probability distribution is clear and random variable $\xi$ is uniformly distributed on $N$ terms, the objective function $f$ can be expressed into a finite-sum form:

$$\min_{x \in \mathbb{R}^d} f(x) := \frac{1}{N} \sum_{i=1}^{N} f_i(x), \tag{1b}$$

where $f_i(x) : \mathbb{R}^d \to \mathbb{R}$ is the $i$-th component function. The optimization of Eqs.(1a) and (1b) is widely encountered in machine learning tasks (Goodfellow et al., 2016; Simonyan & Zisserman, 2014).

To solve this problem given in Eq.(1b), we can compute the gradient of objective function directly with a classic GD (Gradient Descent) algorithm. However, this method suffers from expensive gradient computation for extremely large $N$, and hence people apply stochastic gradient descent method (SGM) to address this issue. Incremental gradient descent (IGD) is a primary version of SGM, where its calculation of gradient proceeds on single component $f_i$ at each iteration, instead of the whole.

As a special case of IGD, SGD (Robbins & Monro, 1951), a fundamental method to train neural networks, always updates parameters by the gradient computed on a mini-batch. There exists an intuitive view that SGD with constant step size (SGD-CS) potentially leads to faster

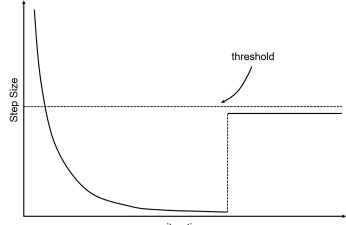

Figure 1: The step size vs. iteration for nonconvex functions: in a neighborhood of the solution, the learning rate can be a constant value below a threshold.

convergence rate. Some studies (Solodov, 1998; Tseng, 1998) show that, under a so-called strong growth condition (SGC), SGD-CS converges to an optimal point faster than SGD with a diminishing step size.

In this work, we study the local convergence (or last-iterate convergence as is called in Jain et al. (2019)) of SGD-CS on nonconvex functions. Note that SGD-CS does not provide a guarantee of converge when starting from any initialization point. Therefore, a useful strategy is to use SGD with a decreasing step size at the beginning, and then switch to SGD-CS in a neighborhood of a minimizer. Fig.1 illustrates the range of the step size of SGD versus the number of iterations.

Our main **theoretical** and **experimental** contributions are as follows.

- We establish **local (or last-iterate) convergence** of SGD with a constant step size (SGD-CS) on nonconvex functions under the interpolation condition. We note that previous results are mostly for strongly convex functions under strong (or weak) growth condition. Our result is much closer to common situations in practice.

- We discover that on linear regression problems with $\ell_2$ regularization, the size of convergent learning rates can be quite large for incremental gradient descent (IGD). Our numerical results show that, within a fairly large range, the larger step size is, the smaller spectral radius is, and the faster the convergence rate is.

- Based on the above observations, we further propose a strategy called SGDL that uses the **SGD with a large initial learning rate** (more than 10 times larger than the learning rate in SGD with momentum), while still being a vanilla SGD. We conduct extensive experiments on various popular deep-learning tasks and models in computer vision, audio recognition and natural language processing. Our results show that the method converges successfully and has a strong generalization performance, and sometimes outperforms its advanced variant (SGDM) and other several popular adaptive methods (e.g., Adam, AdaDelta, etc).

## 2 RELATED WORK

There are many papers on stochastic optimization and we summarize typical ones which are most relevant with our work. The convergence of SGD for over-parameterized models is analyzed in (Vaswani et al., 2018; Mai & Johansson, 2020; Allen-Zhu et al., 2019; Li & Liang, 2018). The power of interpolation is studied in (Ma et al., 2018; Vaswani et al., 2019). The work (Jastrzębski et al., 2017) investigates a large ratio of learning rate to batch size often leads to a wide endpoint. The work (Smith & Topin, 2019) shows a phenomenon called super-convergence which is in contrast to the results in (Bottou et al., 2018). More recently, several new learning rate schedules have proposed for SGD (Loshchilov & Hutter, 2016; Smith, 2017; Agarwal et al., 2017; Carmon et al., 2018).

Adaptive gradient methods are widely in deep learning application. Popular solutions include AdaDelta (Zeiler, 2012), RMSProp(Hinton et al., 2012), Adam(Kingma & Ba, 2014) and so on. Unfortunately, it is believed that the adaptive methods may have a poor empirically performance. For example, Wilson et al. (2017) have observed that Adam hurts generalization performance in comparison to SGD with or without momentum.

The work (Schmidt & Roux, 2013) introduces SGD-CS that attains *linear convergence rate* for *strongly convex* functions under the *strong growth condition* (SGC), and *sublinear convergence rate* for *convex* functions under the SGC. The work (Cevher & Vu, 2018) investigates the *weak growth condition* (WGC), the necessary condition for *linear convergence* of SGD-CS. For a general convex function, the work (Ma et al., 2018) shows that SGD-CS attains linear convergence rate under the *interpolation* property. For a finite sum $f$, the interpolation property means $\nabla f_i(x^*) = 0$ for $i = 1, \ldots, N$, **which is held apparently in over-parameterized DNN.** However, the theoretical analysis of SGD-CS for nonconvex functions is less as mature as in the convex case.

## 3 PRELIMINARIES

### 3.1 NOTATIONS

We denote the minimizer of object function $f$ as $x^*$ and use usual partial ordering for symmetric matrices: $A \succeq B$ means $A - B$ is positive semidefinite; similar for the relations $\preceq, \succ, \prec$. The norm $\|\cdot\|$ either denotes the Euclidean norm for vectors or Frobenius norm for matrices. The Hessian matrix of each component function $f_i(x)$ is denoted as $H_i(x)$. With slight abuse of notation, the Hessian matrix $H_i$ at $x^*$ is denoted as $H_i^*$. We also denote the mean and variance of $\{H_1^*, \ldots, H_N^*\}$ as $H^* \equiv \frac{1}{N} \sum_{i=1}^N H_i^*$ and $\Sigma^* \equiv \frac{1}{N} \sum_{i=1}^N (H_i^*)^2 - (H^*)^2$, and define $a \wedge b = \min(a, b)$. $X \xrightarrow{\mathbb{P}} Y$ means random variable $X$ converges to random variable $Y$ in probability, while $X \xrightarrow{L^1} Y$ means $X$ converges to random variable $Y$ in $L^1$ norm.

### 3.2 ASSUMPTIONS

Our results are under the following mild assumptions:

(A.1) The objective function $f$ has the same structure as given in Eq.(1b).

(A.2) Each component function $f_i$ in Eq.(1b) is twice continously differentiable, with the corresponding Hessian matrix $H_i$ being Lipschitz continuous for constant $L_i$ at the minimizer $x^*$. Without loss of generality, we suppose that

$$\|H_i(x) - H_i(x^*)\| \le L\|x - x^*\|, \quad \text{for } i = 1, \ldots, N, \quad \text{where } L = \max_{i=1,\ldots,N} L_i.$$

(A.3) As in (Zhang et al., 2000, Remark 5.2), we assume the point of interest $x^*$ is a strong minimizer (i.e. $H(x^*) \succ 0$).

Assumption (A.3) restricts that the objective function is essentially strongly convex in a sufficiently small neighborhood of a minimizer. The work (Liu et al., 2020) investigates that the loss function is typically nonconvex in any neighborhood of a minimizer. However, this assumption is not restrictive in the regularization setting.

**Remark 1.** *Even if the Hessian matrix of the point of interest $x^*$ has zero eigenvalues, we can study the $\ell_2$ regularized problem instead, and then the Hessian matrix becomes positive definite. This phenomenon is also validated in our numerical experiment in Section 5, and we add $\ell_2$ regularizer in DNNs in Section 6.*

Based on the above technical assumptions, we focus on the convergence analysis of SGD-CS by discussing the convergence behavior precisely:

$$x_{k+1} = x_k - \alpha \frac{1}{m} \sum_{j=1}^m \nabla f_{\xi_k^{(j)}}(x_k). \tag{2}$$

Next, we introduce the notion of point attraction to help us understand the convergence behavior to Eq.(2).

**Definition 1.** *We say $x^*$ is a point of attraction to Eq.(2) if there is an open ball $\mathcal{N}(x^*, \epsilon)$ of $x^*$, such that for any $x^0 \in \mathcal{N}$, $\{x_k\}$ generated by Eq.(2) all lie in $\mathcal{N}$ and converges to $x^*$ in the $\ell^1$ norm.*

The Ostrowski Theorem says a sufficient condition for $x^*$ to be a point of attraction of the deterministic iteration $x_{k+1} = T(x_k)$ is that the spectral radius of $T'(x)$ is strictly less than one, which further implies the condition that there exists an open ball $\mathcal{N}$ such that once the initial $x_0 \in \mathcal{N}$, the iterates $\{x_k\}$ will converge to $x^*$ with linear rate. We define the strong minimizer $x^*$ as a point of strong attraction following the similar manner.

**Definition 2.** *We say $x^*$ is a point of strong attraction to Eq.(2) if there exists a neighborhood $\mathcal{N}$ of $x^*$, such that for any $x^0 \in \mathcal{N}$, the sequence $\{x_k\}$ generated by Eq.(2) all lie in $\mathcal{N}$ and satisfy*

$$\mathbb{E}\|x^k - x^*\|^2 \le \rho^k \cdot \|x^0 - x^*\|^2, \quad \text{for some } \rho \in [0, 1).$$

The following discussion studies necessary and sufficient conditions for the minimizer $x^*$ being a point of attraction of SGD with the constant step size $\alpha$. However, starting from any initialization point, SGD-CS cannot be guaranteed to find such a local neighborhood around $x^*$, except in some special cases including the loss function studied in Section 5 and Appendix A.2. Therefore, a practical strategy is to use diminishing step size when starting from a random initial point (Allen-Zhu, 2018), and as long as the iteration points belong to a neighborhood of a point of attraction, one can use a constant step size.

## 4 RESULTS FOR NONCONVEX FUNCTIONS

In this section, we rigorously show the necessary condition for the minimizer $x^*$ being a point of attraction, and Theorem 1 provides a sufficient condition for the strong minimizer $x^*$ to be a point of strong attraction with high probability. The detailed proofs are in Appendix.A.1.

### 4.1 NECESSARY CONDITION

**Lemma 1.** *Suppose that the assumptions (A.1) and (A.2) hold, then*

$$\|\nabla f_i(x)\| < L\epsilon^2 + \|H_i^*\|\epsilon \quad for\ i = 1, \ldots, N,$$

*provided that $x \in \mathcal{N}(x^*, \epsilon)$.*

**Theorem 1.** *Suppose that the assumptions (A.1) and (A.2) hold. If the minimizer $x^*$ is a point of attraction of Eq.(2), then the interpolation property is satisfied, i.e.,*

$$\nabla f_i(x^*) = 0 \quad for\ i = 1, \ldots, N.$$

### 4.2 SUFFICIENT CONDITION

#### 4.2.1 SUFFICIENT CONDITION FOR POINTS OF STRONG ATTRACTION

Define the error function $e_i(x)$ as

$$e_i(x) = -H_i^* \cdot (x - x^*) + [\nabla f_i(x) - \nabla f_i(x^*)]. \tag{3}$$

This error function quantifies the residual for the first-order Taylor expansion for $\nabla f_i(x)$ around the point $x^*$ since

$$\nabla f_i(x) = \nabla f_i(x^*) + H_i^*(x - x^*) + e_i(x).$$

By substituting $e_i(x)$ into Eq.(2), we have

$$
\begin{aligned}
x_{k+1} - x^* &= x_k - \alpha \frac{1}{m} \sum_{j=1}^{m} [H_{\xi_k^{(j)}}^* \cdot (x_k - x^*) + e_{\xi_k^{(j)}}(x_k)] - x^* \\
&= \frac{1}{m} \sum_{j=1}^{m} \left[ (I - \alpha H_{\xi_k^{(j)}}^*)(x_k - x^*) - \alpha e_{\xi_k^{(j)}}(x_k) \right].
\end{aligned}
\tag{4}
$$

In addition, our proof for the sufficient condition indicates a bound on the error function.

**Lemma 2.** *Suppose that the assumptions (A.1) and (A.2) hold, and $x^*$ is a local minimizer of $f$. If the interpolation property is satisfied, then the error function defined in Eq.(3) is bounded by:*

$$\|e_i(x)\| \le L\|x - x^*\|^2. \tag{5}$$

Under the assumption that $\{x_k\}$ stay in the local neighborhood, we can show that interpolation property is a sufficient condition for $x^*$ being a point of strong attraction of Eq.(2), provided that the step-size $\alpha$ is sufficiently small.

**Theorem 2.** *Suppose that the assumptions (A.1)-(A.3) hold, then $\{x_k\}$ generated from Eq.(2) all lie in the neighborhood $\mathcal{N}(x^*, \epsilon)$ and $\mathbb{E}[\|x^k - x^*\|^2] \le \rho \mathbb{E}[\|x^{k-1} - x^*\|^2]$, if the radius $\rho$ and the step*

*size $\alpha$ satisfy the following conditions:*

$$\rho \equiv \lambda_{\max}\left(I - 2\alpha H^* + \alpha^2(H^*)^2 + \alpha^2\Sigma^* + \alpha^2 L^2\epsilon^2 I + 2\alpha L\epsilon I\right) < 1, \tag{6a}$$

$$0 < \alpha \le \min_{i=1,\ldots,N}\left\{\frac{1}{\lambda_{\max}(H_i^*)}, \frac{1}{2\lambda_{\max}(H^*)}, \frac{1}{L}\lambda_{\min}(H^*)\right\}, \tag{6b}$$

$$0 < \alpha < 2\lambda_{\min}\left([(H^*)^2 + \Sigma^* + L^2\epsilon^2 I]^{-1/2}(H^* - L\epsilon I)[(H^*)^2 + \Sigma^* + L^2\epsilon^2 I]^{-1/2}\right). \tag{6c}$$

*If $\nabla f_i(x^*) = 0$ for $i = 1, \ldots, N$, then the minimizer $x^*$ is a point of strong attraction of the iteration (2).*

### 4.2.2 STAYING IN THE LOCAL NEIGHBORHOOD

Now we show that $\{x_k\}$ generated from Eq.(2) all stay in the local neighborhood with high probability. The proof follows the idea in (Tan & Vershynin, 2019). In the remaining of this section, we abuse the notation slightly and denote $X_k$ as a random vector in the $k$-th iteration of SGD-CS scheme, and $x_k$ as the realization of $X_k$. Considering the setup of Theorem 2 without assuming that $\{x_k\}$ always lies in the neighborhood $\mathcal{N}(x^*, \epsilon)$, we have the following technical results.

**Lemma 3.** *Let $x_0 \in \mathcal{N}(x^*, \epsilon)$ such that $\|x_0 - x^*\| \le \sqrt{\delta}\epsilon$ for some $\delta \in [0, 1)$. Define the stopping time $\tau = \min\{k : X_k \notin \mathcal{N}(x^*, \epsilon)\}$, then $P\{\tau = \infty\} \ge 1 - \delta$.*

**Remark 2.** *With the setup stated in Lemma 3, the sequence $\{x_k\}$ stay in the local neighborhood $\mathcal{N}(x^*, \epsilon)$ with probability at least $1 - \delta$.*

Indeed, we can show the result in Theorem 2 by relaxing the assumption into that the sequence generated by Eq.(2) stays in the local neighborhood within infinite iterates. Under the setup stated in the Lemma 3, we build the sufficient condition for points of strong attraction with the probabilistic guarantee.

**Theorem 3.** *There exists an event $E$ which holds with probability at least $P(E) \ge 1 - \delta$ and the sequence $\{x_k\}$ generated by Eq (2) satisfies*

$$\mathbb{E}[\|X_k - x^*\|^2 1_E] \le \rho^k\|x_0 - x^*\|^2 \tag{7}$$

**Remark 3.** *It's clear that $x^*$ is a point of attraction if it is a point of strong attraction. Therefore, once the event that the iterates generated from SGD-CS stay in the local neighborhood happens, the interpolation property becomes a necessary and sufficient condition for the strong minimizer $x^*$ to be a point of attraction to Eq.(2).*

## 5 CASE STUDIES ON LARGE LEARNING RATES

This section shows Incremental Gradient Descent (IGD) with a constant step size actually converges into the global optimum point for some special problems, and then some numerical experiments are presented to illustrate this point. We present the results through an example as follows, with more examples appeared in Appendix.A.2.

$$f(x) = \frac{1}{2}\|Ax - b\|^2 + \frac{\mu}{2}\|D^{1/2}x\|^2. \tag{8}$$

Under some mild assumptions (c.f Example 3 in Appendix.A.2), the IGD with constant step size $t$ converges when

$$t \in \left(0, \frac{2N}{\rho(\mu D + A^\mathrm{T} A)}\right).$$

where $\rho$ is the spectral radius.

The problem in Eq.(8) can be extended into general nonlinear least squares:

$$f(x) = \frac{1}{2}\sum_{i=1}^{N}r_i^2(x) = \frac{1}{2}R(x)^T R(x).$$

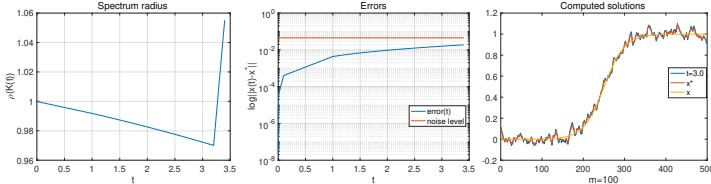

Figure 2: Left: the range of spectral radius where IGD converges, it can be seen there are a wide range $t$ for convergence. Middle: the Log error $\|x - \hat{x}\|_2$. The error line in blue grows slowly after an initial rapid rise. This implies that in the presence of noise, large t-values can still calculate $x$ close to $x^*$. Right: calculated solution of a large radius (3.0),the computed values of $x(t)$ do not significantly deviate from the noisy data $\hat{x}$ within the large $t$-value.

A sufficient condition for the convergence of IGD is

$$t \in \left(0, \frac{1}{\lambda_{\max}(\mathbb{E}[H_i(x^*)])}\right),$$

where $\mathbb{E}[H_i(x^*)] \equiv \frac{1}{N}\sum_i H_i(x^*)$, and $H_i(\cdot)$ denotes the Hessian matrix for the $i$-th component function $r_i(\cdot)$.

We conduct a numerical simulation for the regularized least squares problem in Eq.(8). Let training data $x$ be generated via $x = \frac{1}{1+e^z} + N(0, \sigma^2)$ where $z$ is uniformly sampled from $(0, 10)$. The minimizer $x^*$ does not necessarily represent the best solution available due to noise, so all approximate solutions within the same noise level should be equally good in fact.

The experimental results demonstrate that with large step sizes, IGD still converge and have a decent performance, which meets our expectations. After the first sharp rise, the step size can be updated insensitive to the error. In our next section, we will implement vanilla SGD with a large initial learning rate and use this optimizer for large scale problems.

## 6 EXPERIMENTS

We show the empirical results of different models to compare our SGDL with other popular optimization methods, including SGD, SGDM, Adam, and AdaDelta. We focus on the CIFAR10 and CIFAR100 image classification task (Krizhevsky et al., 2009), with the downsampled variant of ImageNet named as ImageNet32 (Chrabaszcz et al., 2017), the Speech Commands Dataset for audio recognition(Warden, 2018), and the language modeling task on Penn Treebank(Marcus et al., 1993).

### 6.1 CIFAR10 AND CIFAR100

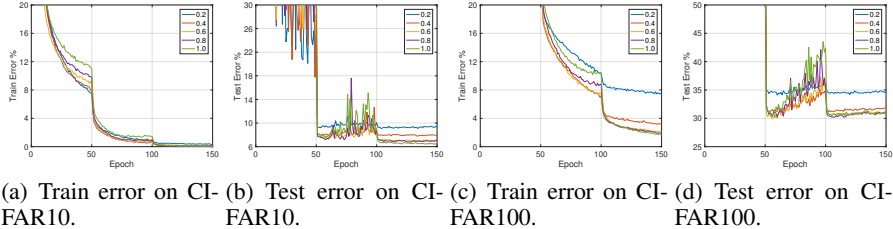

(a) Train error on CIFAR10.  (b) Test error on CIFAR10.  (c) Train error on CIFAR100.  (d) Test error on CIFAR100.

Figure 3: Different learning rate for the vanilla SGD in the ResNet56 model on CIFAR10 and CIFAR100. Models are trained with learning rate from 0.2 to 1.

The first experiment is carried out on the CIFAR10 and CIFAR100 datasets, which are standard image collection with 10 and 100 classes respectively. After the preprocessing in Appendix.A.3.1, we train VGG (Simonyan & Zisserman, 2014), ResNet (He et al., 2016) and DenseNet(Huang et al., 2017) for up to 150 epochs and minibatch size to 256. Additional, for all optimizers without

AdaDelta, we use an annealing strategy that the learning rate is lowered by 10 times at epoch 50 and 100. For ResNet experiments, we select a ResNet network with 56 layers and 120 layers respectively. For VGG, we use VGG16. For DenseNet, we use a DenseNet with 100 layers and growth rate $k = 12$. We first run a ResNet56 model on CIFAR10 and CIFAR100 with different learning rates. From the results show in Fig.3 we can see a small learning rate like 0.2, 0.4 has a relatively poor performance on the test set.

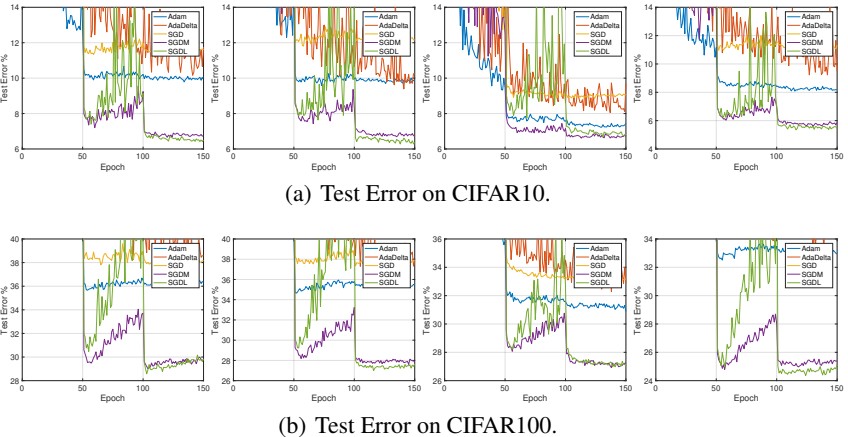

(a) Test Error on CIFAR10.

(b) Test Error on CIFAR100.

Figure 4: The performance of several popular models on CIFAR10 and CIFAR100. Figures from left to right are for ResNet56, ResNet110,VGG16 and DenseNet, respectively.

We compare different optimization method including SGD,SGDM,Adam,AdaDelta, and shows the performance in Fig.4. We can see that our method still works fairly well and has the same or even better performance in comparison to SGDM and other optimizers.

## 6.2 IMAGENET32

The second experiment is carried out on ImageNet32 dataset (Deng et al., 2009) with more than a million images in 1000 classes, out of which 50000 images are used as a testing set. Each image has $32 \times 32$ pixels. The objective is to train an image classifier. We apply a similar train strategy as on the CIFAR10 and CIFAR100 datasets in Section 6.1, except that the epoch budget is down to 120 and the learning rate is shrunk to one-tenth of its current value every 30 epochs.

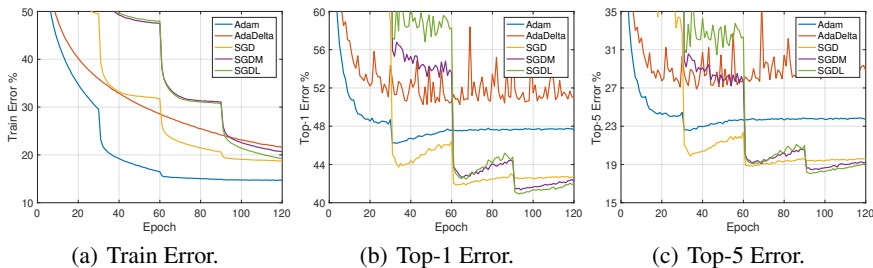

(a) Train Error.  (b) Top-1 Error.  (c) Top-5 Error.

Figure 5: Left: Train error for ResNet56 on ImageNet32; Middle: Top-1 error; Right: Top-5 error.

Our experiments is only focus on ResNet56 due to computational resource limitation. To adjust the difficulty on this dataset, we increase the model capacity for the previously used model by quadrupling the number of channels on each convolution layer. From the results on Fig.5, we find our method outperforms SGDM slightly about 0.5% for the best accuracy. More interestingly, SGD also has a good performance, while two adaptive methods both work just passably.

## 6.3  AUDIO RECOGNITION

The third experiment is carried out on the Speech Commands Dataset, which consists of recordings from thousands of different people in uncontrolled recording conditions. Each sample is represented as a 16000-dimension vector. Each recording is one second in length. We divide the data into a training set and a testing set.

With the dataset, we train a 2-layered neural net with 20 channels, a 2D dropout layer and 2 full-connection layers with 1,000 hidden nodes. More specific hpyerparameter settings is in Appendix.A.3.2. From the results in Fig.6, we can see that SGDL has quite competitive effect and outperforms vanilla SGD exceedingly.

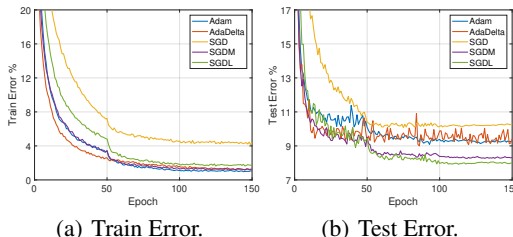

(a) Train Error.  (b) Test Error.

Figure 6:  LeNet5 on Speech Commands Dataset.

## 6.4  LSTM LANGUAGE MODEL

The fourth experiment is carried out on the standard language modeling task Penn Treebank (PTB) dataset. The learning rate is selected and optimized for several years and the current state-of-the-art results supported our perspective about large learning rate (Merity et al., 2017a). We focus to compare the effect of SGDL, SGDM and other optimizers on top of a state-of-the-art $\{2, 3\}$- LSTM training recipe with some training tricks (Merity et al., 2017b;a; Inan et al., 2016; Gal & Ghahramani, 2016). The details on LSTMs are in Appendix.A.3.3.

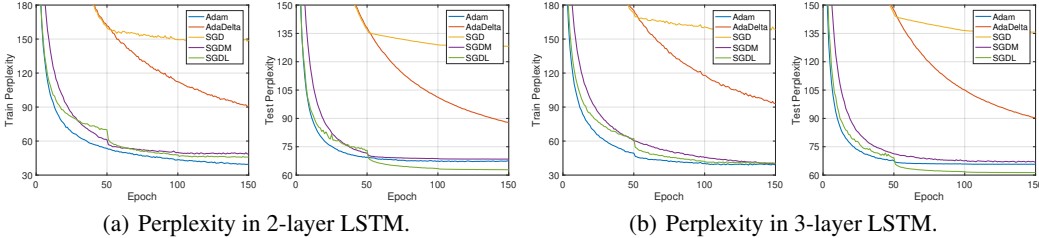

(a) Perplexity in 2-layer LSTM.  (b) Perplexity in 3-layer LSTM.

Figure 7:  LSTM on Penn Treebank Dataset. For Fig. (a) and Fig. (b), the left shows Train Perplexity and the right shows Test Perplexity.

From the result in Fig.7, we evaluate the performance by perplexity metric. We find that Adam is the fastest on the initial progress, but its final performance is worse than our SGDL. We also investigate AdaDelta has terrible performance, while SGDM has almost same curves with ADAM.

## 6.5  DISCUSSION

In Sections 6.1-6.4, we have observed that if vanilla SGD has the same learning rate as SGDM, it tends to a poor generalization performance in some tasks and this is one of the reasons why SGD is usually shelved for training models nowadays. Our experiments show that a larger initial learning rate increases the test accuracy on Fig.3. Further more, it can be seen that SGDL has an admirable performance in our experiments and outperforms SGDM in some tasks. For various deep neural networks such as ResNet,DenseNet,VGG, our SGDL works well, which is not affected by abundance layers or the existence of the residual mechanism. The experimental results have provided evidence that SGD with large learning rate is a good alternative to SGD with momentum in some practical tasks.

## 7 CONCLUSION

We provide a rigorous proof on the local convergence behavior of SGD-CS on smooth and nonconvex functions. Motivated by numerical experiments on IGD, we recommend a vanilla SGD with large learning rate (SGDL) for training neural networks. Extensive evaluations have been carried out in deep learning tasks (CV,Speech,NLP) with popular neural network architectures. The results have demonstrated the smooth convergence and effective generalization performance of SGDL. For our future work, we will analyze the effect from more training strategies such as batch normalization about convergence. We will also analyze the pros and cons in SGD and its variants (SGDM) in our further work.

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

## A    APPENDIX

### A.1    PROOFS FOR SECTION 4

*Proof of Lemma 1.* For $i = 1, \ldots, N$, by Lipschitz continuity of $H_i$ at $x^*$, for any $y \in \mathcal{N}(x^*, \epsilon)$,

$$\|H_i(y) - H_i^*\| \leq L\|y - x^*\| < L\epsilon,$$

and therefore $\|H_i(y)\| \leq \|H_i(y) - H_i^*\| + \|H_i^*\| < L\epsilon + \|H_i^*\|$. For fixed $x \in \mathcal{N}(x^*, \epsilon)$, define $g_i(t) = \nabla f_i(x^* + t(x - x^*))$, and therefore $g_i'(t) = (x - x^*)^{\mathrm{T}} H_i(x^* + t(x - x^*))$. By the fundamental theorem of calculus,

$$
\begin{aligned}
\|\nabla f_i(x) - \nabla f_i(x^*)\| = \|g_i(1) - g_i(0)\| &= \left\| \int_0^1 g_i'(t)\, \mathrm{d}t \right\| \\
&\leq \int_0^1 \|g_i'(t)\|\, \mathrm{d}t = \int_0^1 \|(x - x^*)^{\mathrm{T}} H_i(x^* + t(x - x^*))\| \\
&\leq \int_0^1 \|x - x^*\| \|H_i(x^* + t(x - x^*))\|\, \mathrm{d}t \\
&< \int_0^1 \|x - x^*\|(L\epsilon + \|H_i^*\|)\, \mathrm{d}t \\
&< L\epsilon^2 + \|H_i^*\|\epsilon.
\end{aligned}
$$

$\square$

*Proof of Theorem 1.* Being a point of attraction, the sequence $\{x_k\}$ satisfies the Cauchy criteria:

$$\lim_{N \to \infty} \sup_{m,n \geq N} \mathbb{E}\|x_m - x_n\| = 0.$$

Then $\lim_{k \to \infty} \mathbb{E}\|x_{k+1} - x_k\| = 0$. Substituting $x_{k+1}$ with the iteration (2),

$$
\begin{aligned}
0 &= \lim_{k \to \infty} \mathbb{E}\alpha \left\| \frac{1}{m} \sum_{j=1}^m \nabla f_{\xi_k^{(j)}}(x_k) \right\| \\
&= \alpha \lim_{k \to \infty} \mathbb{E} \left\{ \mathbb{E}\left[ \left\| \frac{1}{m} \sum_{j=1}^m \nabla f_{\xi_k^{(j)}}(x_k) \right\| \Big| x_k \right] \right\} \\
&= \alpha \lim_{k \to \infty} \mathbb{E} \sum_{\ell_{1:m} \in \{1, \ldots, N\}^m} \frac{1}{N^m} \cdot \left\| \frac{1}{m} \sum_{j=1}^m \nabla f_{\ell_j}(x_k) \right\|.
\end{aligned}
$$

Since $\|\frac{1}{m}\sum_{j=1}^m \nabla f_{\ell_j}(x_k)\| \geq 0$ for $\ell_{1:m} \in \{1, \ldots, N\}^m$ and all iterates $k$,

$$\lim_{k \to \infty} \mathbb{E}\left\| \frac{1}{m} \sum_{j=1}^m \nabla f_{\ell_j}(x_k) \right\| = 0, \quad \text{for any } \ell_{1:m} \in \{1, \ldots, N\}^m.$$

In particular, when $\ell_1 = \cdots = \ell_m = i$ for fixed $i = 1, \ldots, N$,

$$\lim_{k \to \infty} \mathbb{E}\|\nabla f_i(x_k)\| = 0. \tag{9}$$

For fixed $i = 1, \ldots, N$, the hyphothesis of the bounded convergence theorem (Durrett, 2019, Theorem 1.5.3) is satisfied:

- $\|\nabla f_i(x)\| \leq L\epsilon^2 + \|H_i^*\|\epsilon$ for any $x \in \mathcal{N}(x^*, \epsilon)$, following the result in Lemma (1).

- $\|\nabla f_i(x_k)\| \xrightarrow{\mathbb{P}} \|\nabla f_i(x^*)\|$: $x_k \xrightarrow{L^1} x^*$ implies $\|\nabla f_i(x_k)\| \xrightarrow{L^1} \|\nabla f_i(x^*)\|$. By comparison theorem for convergence, $\|\nabla f_i(x_k)\| \xrightarrow{\mathbb{P}} \|\nabla f(x^*)\|$.

Applying the bounded convergence theorem on (9) over the region $\mathcal{N}(x^*, \epsilon)$ gives

$$\mathbb{E} \lim_{k \to \infty} \|\nabla f_i(x_k)\| = 0 \implies \mathbb{E}\|\nabla f_i(x^*)\| = \|\nabla f_i(x^*)\| = 0, \quad \text{for } i = 1, \ldots, N,$$

which implies the desired result. $\qquad \square$

*Proof of Lemma 2.* By the interpolation property and definition of $e_i(x)$ for $i = 1, \ldots, N$,

$$\begin{aligned}
e_i(x) &= -H_i^* \cdot (x - x^*) + \nabla f_i(x) \\
&= -H_i^* \cdot (x - x^*) + \nabla f_i(x^* + (x - x^*)) \\
&= -H_i^* \cdot (x - x^*) + [\nabla f_i(x^*) + H_i(x^* + t(x - x^*)) \cdot (x - x^*)] \quad \text{for some } t \in [0, 1]
\end{aligned} \tag{10a}$$

$$= [H_i(x^* + t(x - x^*)) - H_i^*] \cdot (x - x^*), \tag{10b}$$

where (10a) is by the Taylor expansion on $\nabla f_i(x)$, and (10b) is because $\nabla f_i(x^*) = 0$. By Cauchy-schwarz inequality,

$$\begin{aligned}
\|e_i(x)\| &\leq \|H_i(x^* + t(x - x^*)) - H_i^*\|\|x - x^*\| \\
&\leq L\|t(x - x^*)\|\|x - x^*\| \\
&\leq L\|x - x^*\|^2,
\end{aligned} \tag{10c}$$

where (10c) is by Lipschitz continuity of $H_i$ at $x^*$. $\qquad \square$

*Proof of Theorem 2.* Since the step size $\alpha \leq \min_{i \in \{1, \ldots, N\}} \frac{1}{\lambda_{\max}(H_i^*)}$, the term

$$0 \preceq I - \alpha H_i^* \preceq I, \quad \text{for } i = 1, \ldots, N.$$

Substituting (4) into the term $\|x_{k+1} - x^*\|^2$ gives

$$\begin{aligned}
\|x_{k+1} - x^*\|^2 &= \left\| \frac{1}{m} \sum_{j=1}^{m} \left[ (I - \alpha H_{\xi_k^{(j)}}^*)(x_k - x^*) - \alpha e_{\xi_k^{(j)}}(x_k) \right] \right\|^2 \\
&\leq \max_{j \in \{1, \ldots, m\}} (x_k - x^*)^{\mathrm{T}}(I - \alpha H_{\xi_k^{(j)}}^*)^2(x_k - x^*) \\
&\quad + \alpha^2 \|e_{\xi_k^{(j)}}(x_k)\|^2 - 2\alpha(x_k - x^*)^{\mathrm{T}}(I - \alpha H_{\xi_k^{(j)}}^*)e_{\xi_k^{(j)}}(x_k) \\[6pt]
&\leq \max_{j \in \{1, \ldots, m\}} (x_k - x^*)^{\mathrm{T}}(I - \alpha H_{\xi_k^{(j)}}^*)^2(x_k - x^*) \\
&\quad + \alpha^2 \|e_{\xi_k^{(j)}}(x_k)\|^2 + 2\alpha\|x_k - x^*\|\|I - \alpha H_{\xi_k^{(j)}}^*\|\|e_{\xi_k^{(j)}}(x_k)\| \\[6pt]
&\leq \max_{j \in \{1, \ldots, m\}} (x_k - x^*)^{\mathrm{T}}(I - \alpha H_{\xi_k^{(j)}}^*)^2(x_k - x^*) \\
&\quad + \alpha^2 L^2 \|x_k - x^*\|^4 + 2\alpha L \|I - \alpha H_{\xi^{(k)}}^*\|\|x_k - x^*\|^3 \\[6pt]
&= \max_{j \in \{1, \ldots, m\}} (x_k - x^*)^{\mathrm{T}} \\
&\quad \left( (I - \alpha H_{\xi_k^{(j)}}^*)^2 + \alpha^2 L^2 \|x_k - x^*\|^2 + 2\alpha L\|x_k - x^*\|\|I - \alpha H_{\xi_k^{(j)}}^*\| \right)(x_k - x^*) \\[6pt]
&\leq \max_{j \in \{1, \ldots, m\}} (x_k - x^*)^{\mathrm{T}} \left( (I - \alpha H_{\xi_k^{(j)}}^*)^2 + \alpha^2 L^2 \epsilon^2 + 2\alpha L \epsilon \right)(x_k - x^*)
\end{aligned}$$

with labels (11a), (11b), (11c), (11d) respectively on the inequality lines.

where (11a) follows from the inequality $\|\frac{1}{D}\sum_{i=1}^{d}a_i\|^2 \leq \max_{i\in\{1,...,d\}}\|a_i\|^2$; (11b) follows from the Cauchy-Schwarz inequality on the last term; (11c) follows from Lemma (2); (11d) is because that the sequence $\{x_k\} \in \mathcal{N}(x^*, \epsilon)$, and that $\|I - \alpha H^*_{\xi^{(k)}}\| \leq 1$.

Therefore, taking conditional expectation both sides given $x_k$ implies

$$\mathbb{E}[\|x_{k+1} - x^*\|^2 \mid x_k] \leq \max_{j\in\{1,...,m\}}\left[(x_k - x^*)^{\mathrm{T}}\right.$$
$$\left(\mathbb{E}_{\xi_k^{(j)}}[(I - \alpha H^*_{\xi^{(k)}})^2] + \alpha^2 L^2 \epsilon^2 + 2\alpha L\epsilon\right) \tag{12}$$
$$\left.(x_k - x^*)\right]$$

The expectation term $\mathbb{E}_{\xi_k^{(j)}}\{(I - \alpha H^*_{\xi^{(k)}})^2\}$ can be simplified as follows:

$$\mathbb{E}_{\xi_k^{(j)}}[(I - \alpha H^*_{\xi_k^{(j)}})^2] = \mathbb{E}_{\xi_k^{(j)}}\left[I - 2\alpha H^*_{\xi_k^{(j)}} + \alpha^2(H^*_{\xi_k^{(j)}})^2\right]$$
$$= I - 2\alpha H^* + \alpha^2\mathbb{E}_{\xi_k^{(j)}}[(H^*_{\xi_k^{(j)}})^2]$$
$$= I - 2\alpha H^* + \alpha^2\frac{1}{N}\sum_{i=1}^{N}(H^*_i)^2 \tag{13}$$
$$= I - 2\alpha H^* + \alpha^2(H^*)^2 + \alpha^2\left(\frac{1}{N}\sum_{i=1}^{N}(H^*_i)^2 - (H^*)^2\right)$$
$$= I - 2\alpha H^* + \alpha^2(H^*)^2 + \alpha^2\Sigma^*.$$

Combining (12) and (13) gives an upper bound on $\mathbb{E}[\|x_{k+1} - x^*\|^2 \mid x_k]$:

$$\mathbb{E}[\|x_{k+1} - x^*\|^2 \mid x_k] \leq (x_k - x^*)^{\mathrm{T}}\left(I - 2\alpha H^* + \alpha^2(H^*)^2 + \alpha^2\Sigma^* + \alpha^2 L^2\epsilon^2 I + 2\alpha L\epsilon I\right)(x_k - x^*)$$

Since $\alpha$ and $\epsilon$ are sufficiently small such that

$$0 \leq \rho \equiv \lambda_{\max}\left(I - 2\alpha H^* + \alpha^2(H^*)^2 + \alpha^2\Sigma^* + \alpha^2 L^2\epsilon^2 I + 2\alpha L\epsilon I\right) < 1,$$

the conditional expectation is further upper bounded by

$$\mathbb{E}[\|x_{k+1} - x^*\|^2 \mid x_k] \leq \rho\|x_k - x^*\|^2.$$

Therefore,
$$\mathbb{E}[\|x_{k+1} - x^*\|^2] = \mathbb{E}\left\{\mathbb{E}[\|x_{k+1} - x^*\|^2 \mid x_k]\right\} \leq \rho\mathbb{E}[\|x_k - x^*\|^2]. \tag{14}$$

Furtheremore,
$$\mathbb{E}[\|x^k - x^*\|^2] \leq \rho\mathbb{E}[\|x^{k-1} - x^*\|^2] \leq \cdots \leq \rho^k\|x^0 - x^*\|.$$

By (6b),
$$H^* \succ L\epsilon I, \tag{15}$$
$$I - 2\alpha H^* \succeq 0. \tag{16}$$

Since $H^* - L\epsilon I \succ 0$ and by (6c),
$$\alpha((H^*)^2 + \Sigma^*) + \alpha L^2\epsilon^2 I \prec 2H^* - 2L\epsilon I. \tag{17}$$

Combining (16) and (17) gives
$$0 \preceq I - 2\alpha H^* + \alpha^2(H^*)^2 + \alpha^2\Sigma^* + \alpha^2 L^2\epsilon^2 I + 2\alpha L\epsilon I \prec I.$$

$\square$

*Proof of Lemma 3.* Let $\mathcal{F}_k$ denote the $\sigma$-algebra generated by the first $k$ SGD-CS random vectors $\xi_1^{(1:m)}, \ldots, \xi_k^{(1:m)}$. Construct $Z_k = \frac{\|X_{\tau \wedge k} - x^*\|^2}{\rho^{\tau \wedge k}}$. Firstly we show that $Z_k$ is a supermatingale:

$$
\mathbb{E}[Z_{k+1} \mid \mathcal{F}_k] = \mathbb{E}\left[\frac{\|X_{\tau \wedge (k+1)} - x^*\|^2}{\rho^{\tau \wedge (k+1)}} 1_{\tau \leq k} \bigg| \mathcal{F}_k\right] + \mathbb{E}\left[\frac{\|X_{\tau \wedge (k+1)} - x^*\|^2}{\rho^{\tau \wedge (k+1)}} 1_{\tau > k} \bigg| \mathcal{F}_k\right]
$$

$$
= \mathbb{E}\left[\frac{\|X_{\tau \wedge k} - x^*\|^2}{\rho^{\tau \wedge k}} 1_{\tau \leq k} \bigg| \mathcal{F}_k\right] + \mathbb{E}\left[\frac{\|X_{k+1} - x^*\|^2}{\rho^{k+1}} 1_{\tau > k} \bigg| \mathcal{F}_k\right]
$$

$$
\overset{(i)}{=} Z_k 1_{\tau \leq k} + \frac{1}{\rho^{k+1}} \mathbb{E}\left[\|X_{k+1} - x^*\|^2 1_{\tau > k} \big| \mathcal{F}_k\right]
$$

$$
\overset{(ii)}{\leq} Z_k 1_{\tau \leq k} + \frac{1}{\rho^{k+1}} \rho \mathbb{E}\left[\|X_k - x^*\|^2 1_{\tau > k} \big| \mathcal{F}_k\right]
$$

$$
= Z_k 1_{\tau \leq k} + Z_k 1_{\tau > k} = Z_k,
$$

where (i) is because that $\frac{\|X_{\tau \wedge k} - x^*\|^2}{\rho^{\tau \wedge k}} 1_{\tau \leq k}$ is measurable with respect to $\mathcal{F}_k$; (ii) is by applying (14) in Theorem (2). As a result,

$$
Z_0 \geq \mathbb{E}[Z_k \mid \mathcal{F}_0] \geq \mathbb{E}\left[\frac{\|X_{\tau \wedge k} - x^*\|^2}{\rho^{\tau \wedge k}} 1_{k \geq \tau} \bigg| \mathcal{F}_0\right] \geq \mathbb{E}\left[\frac{\|X_\tau - x^*\|^2}{\rho^\tau} 1_{k \geq \tau} \bigg| \mathcal{F}_0\right].
$$

By definition of the stopping time, $\|X_\tau - x^*\|^2 \geq \epsilon^2$; and the term $Z_0 := \|x_0 - x^*\|^2 \leq \delta \epsilon^2$ since $\|x_0 - x^*\| \leq \sqrt{\delta} \epsilon$. Substituting these two relations into the inequality above gives

$$
\delta \epsilon^2 \geq \mathbb{E}\left[\frac{\epsilon^2}{\rho^\tau} 1_{k \geq \tau} \mid \mathcal{F}_0\right] \implies \delta \geq \mathbb{E}\left[\frac{1_{k \geq \tau}}{\rho^\tau} \bigg| \mathcal{F}_0\right] \geq \mathbb{E}\left[1_{k \geq \tau} | \mathcal{F}_0\right] \geq P\{k \geq \tau\}, \forall k.
$$

Or after a rearrangement, $\delta \geq P(\tau < \infty)$, which implies $P(\tau = \infty) \geq 1 - \delta$.

Furthermore,
$$
\mathbb{E}[\|X_k - x^*\|^2 1_{\tau = \infty}] = \mathbb{E}[\|X_k - x^*\|^2 \mid \tau = \infty] P(\tau = \infty) \geq (1 - \delta) \mathbb{E}[\|X_k - x^*\|^2 \mid \tau = \infty].
$$
(18)

Apply Theorem (2) to bound $\mathbb{E}[\|X_k - x^*\|^2 1_{\tau = \infty}]$:
$$
\mathbb{E}[\|X_k - x^*\|^2 1_{\tau = \infty}] \leq \rho^k \|x_0 - x^*\|^2.
$$
(19)

We now estimate $\mathbb{E}[\|X_k - x^*\|^2 \mid \tau = \infty]$ by utilizing (18) and (19):
$$
\mathbb{E}[\|X_k - x^*\|^2 \mid \tau = \infty] \leq \frac{\rho^k}{1 - \delta} \|x_0 - x^*\|^2.
$$

$\square$

*Proof of Theorem 3.* Define the event $E = \{\tau = \infty\}$. It suffices to show the relation (7). By direct calculation,

$$
\mathbb{E}[\|X_{k+1} - x^*\|_2^2 1_{\tau > k+1} \mid X_k = x_k] \leq \mathbb{E}[\|X_{k+1} - x^*\|_2^2 1_{\tau > k} \mid X_k = x_k]
$$

$$
= \mathbb{E}[\|X_{k+1} - x^*\|_2^2 1_{\tau > k} \mid X_k = x_k, \mathcal{F}_k]
$$

$$
= \mathbb{E}[\|X_{k+1} - x^*\|_2^2 \mid X_k = x_k, \mathcal{F}_k] 1_{\tau > k}
$$

$$
\leq \rho \|x_k - x^*\|^2 1_{\tau > k}
$$

where the last inequality follows from (14). As a result,
$$
\mathbb{E}[\|X_{k+1} - x^*\|_2^2 1_{\tau > k+1}] = \mathbb{E}\left\{\mathbb{E}[\|X_{k+1} - x^*\|_2^2 1_{\tau > k+1} \mid X_k]\right\}
$$

$$
\leq \rho \mathbb{E}[\|X_k - x^*\|^2 1_{\tau > k}]
$$

Inductively, $\mathbb{E}[\|X_k - x^*\|^2 1_{\tau > k}] \leq \rho^k \|x_0 - x^*\|^2$. Therefore,
$$
\mathbb{E}[\|X_k - x^*\|^2 1_E] = \mathbb{E}[\|X_k - x^*\|^2 1_{\tau = \infty}] \leq \mathbb{E}[\|X_k - x^*\|^2 1_{\tau > k}] \leq \rho^k \|x_0 - x^*\|^2,
$$
which completes the proof. $\square$

*Proof of Remark 2.* Conditioned on the event $\{\tau = \infty\}$, we have
$$
P(\|X_k - x^*\|^2 > \epsilon^2 \mid \tau = \infty) = 0.
$$
It follows that
$$
P(\|X_k - x^*\|^2 \leq \epsilon^2) \geq P(\|X_k - x^*\|^2 \leq \epsilon^2 \mid \tau = \infty) P(\tau = \infty) \geq (1 - \delta).
$$

$\square$

### A.2 EXAMPLES

**Example 1.** *(**Quadratic Functions**) Consider minimizing for the objective function $f(x) = \sum_{i=1}^{N} x_i^2 \equiv x^{\mathrm{T}} x$, where $x \in \mathbb{R}^{N \times 1}$, and $f_i = x_i^2, i \in [N]$. Then the gradient of each component $f_i$ is given by*

$$\nabla f_i(x) = 2 E_i \cdot x,$$

*where $E_i \in \mathbb{R}^{N \times N}$ is a all-zero matrix except $E_{i,i} = 1$. Then the end-to-end $N$ runs of IGD update with step size $t$ can be expressed as a matrix compact form:*

$$x^{new} = K_N(t) x^{old},$$

*where*

$$K_N(t) = \prod_{i=1}^{N} \left( I - 2t E_i \right) = \mathrm{diag}(1 - 2t, 1 - 2t, \ldots, 1 - 2t). \tag{20}$$

*Applying the basic knowledge in linear algegra, $\rho(K_N(t)) = |1 - 2t|$. As long as $t \in (0, 1)$, the IGD will converge from any initial point.*

**Example 2.** *(**Standard Least Squares Problem**) Consider the standard un-determined least squares problem*

$$f(x) = \frac{1}{2} \|Ax - b\|^2, \quad A \in \mathbb{R}^{m \times N}, N < n.$$

*Similarly, the end-to-end $N$ runs of IGD update with step size $t$ forms the linear system*

$$x^{new} = K_N(t) x^{old} + tc(t),$$

*with*

$$K_0(t) \equiv I, \quad K_j(t) = \prod_{i=1}^{j} \left( I - t a_i a_i^{\mathrm{T}} \right), \quad j = 1, 2, \ldots, N,$$

*and $c(t) = \sum_{j=1}^{N} b_j K_{j-1}(t) a_j$. The sufficient condition for convergence is $\lambda_{\max}(K_N(t)) < 1$, and the necessary condition is $\lambda_{\max}(K_N(t)) \leq 1$. In this situation, we can assert that $\lambda_{\max}(K_N(t)) \geq 1$: we can pick $x_0 \in \mathbb{R}^n \setminus \{0\}$ so that $Ax_0 = 0$, which implies $(K_N(t)) x_0 = x_0$. This means that 1 is an eigenvalue of $K_N(t)$, i.e., $\lambda_{\max}(K_N(t)) \geq 1$. This example also suggests that regularization helps with the convergence in optimization.*

**Example 3.** *(**Regularized Least Squares Problem**) Consider the regularized least squares objective function*

$$f(x) = \frac{1}{2} \|Ax - b\|^2 + \frac{\mu}{2} \|D^{1/2} x\|^2.$$

*It can be shown that the end-to-end $N$ iteration of IGD with step size $t$ can be expressed as a compact matrix form:*

$$x^{new} = K_N(t) x^{old} + tc(t), \tag{21a}$$

$$K_0(t) \equiv I, \quad K_j(t) = \prod_{i=1}^{j} \left( I - t a_i a_i^{\mathrm{T}} - t \frac{\mu}{m} D \right), \quad j = 1, \ldots, N, \tag{21b}$$

$$c(t) = \sum_{j=1}^{m} b_j K_{j-1}(t) a_j. \tag{21c}$$

*The necessary and sufficient conditions for the convergence of IGD suffice to characterize the condition $\rho(t) \equiv \rho(K_N(t)) < 1$. We need to make use of the famous conjecture about matrix AM-GM inequality:*

*For any positive semi-definite matrix $P_1, P_2, \ldots, P_n \in \mathbb{R}^{n \times n}$,*

$$\frac{1}{n!} \sum_{\sigma = (\sigma_1, \ldots, \sigma_n) \in \Gamma} P_{\sigma_1} P_{\sigma_2} \cdots P_{\sigma_n} \preceq \left( \frac{1}{n} \sum_i A_i \right)^n.$$

We make a reasonable assumption that the products within $K_N(t)$ are commutative, then applying this conjecture gives

$$K_N(t) \preceq \left[ \frac{1}{N} \sum_{i=1}^{N} \left( I - ta_i a_i^{\mathrm{T}} - t\frac{\mu}{N}D \right) \right]^N = \left[ I - \frac{t}{N}[\mu D + A^{\mathrm{T}}A] \right]^N$$

Hence, a sufficient condition for the convergence would be:

$$\rho \left( \left[ I - \frac{t}{N}[\mu D + A^{\mathrm{T}}A] \right]^N \right) < 1.$$

Or equivalently, we need to pick $t$ such that, for any eigenvalue $\lambda$ of the matrix $\mu D + A^{\mathrm{T}}A$, we have

$$\left( 1 - \frac{t\lambda}{N} \right)^N < 1 \Longleftrightarrow t \in \left( 0, \frac{2N}{\rho(\mu D + A^{\mathrm{T}}A)} \right).$$

**Example 4.** *(**General Nonlinear Least Squares Problem**) Consider the general nonlinear least squares problem:*

$$f(x) = \frac{1}{2} \sum_{i=1}^{N} r_i^2(x) = \frac{1}{2} R(x)^T R(x).$$

*By assuming that $r_i(x)$ are twice continuously differentiable for all $i \in [N]$, the end-to-end $N$ iteration of IGD with step size $t$ can be safely approximated as*

$$x^{new} = K_N(t)x^{old} + tc(t) + o(\|x^0 - x^*\|^2), \tag{22a}$$

$$K_0(t) \equiv I, \quad K_j(t) = \prod_{i=1}^{j} \left( I - tH_i(x^*) \right), \quad j = 1, \ldots, N, \tag{22b}$$

$$c(t) = \sum_{j=1}^{m} K_{j-1}(t)H_j(x^*)x^*. \tag{22c}$$

*It is reasonable to assume that $H_i(x^*)$ is full rank for all $i$, since adding small regularization terms can resolve the rank deficiency issue. Applying the conjecture again, we imply that*

$$K_N(t) \preceq \left[ I - \frac{t}{N} \sum_i H_i(x^*) \right]^N$$

*Therefore, a sufficient condition for the convergence of IGD would be*

$$t \in \left( 0, \frac{1}{\lambda_{\max}(\mathbb{E}[H_i(x^*)])} \right), \quad where \ \mathbb{E}[H_i(x^*)] \equiv \frac{1}{N} \sum_i H_i(x^*).$$

### A.3 SUPPLEMENT TO EXEPERIEMENTS

#### A.3.1 IMAGE CLASSIFICATION

We normalize data and then augment them by horizontal flips and random crops from the image padded by 4 pixels, filling missing pixels with reflections of the original image. We adopt Kaiming initialization(He et al., 2015), BN (Ioffe & Szegedy, 2015) but without dropout. The models are trained for up to 150 epochs and minibatch size to 256. We select the best learning rate from Table.1 in Appendix.A.3.1 for SGD(M), and for SGDL, the learning rate is more than 10 times larger than the best SGDM $lr$, in our paper, we select it from $\{1.0, 1.1, 1.2, 1.3\}$, more specific parameter settings are as follows. Additional, for all optimizers without AdaDelta, we use an annealing strategy to divide learning rate by 10 on 50 and 100 epochs.

The figures show results for ResNet110,VGG,DenseNet on CIFAR10 and CIFAR100. For each figure, the specific learning rate is given for SGDL. On **CIFAR10**, SGDL learning rate is 1.0 for ResNet 56, 1.1 for DenseNet and 1.2 for ResNet110 and VGG16, while on **CIFAR100**, learning rate is 1.0 for VGG16 and 1.3 for ResNet 56, ResNet110 and DenseNet on SGDL. For SGDM, the learning rate is 0.1 for most experiments, while it is 0.05 for VGG16 on CIFAR10. Finally, 0.001 is the most suitable learning rate for ADAM. On **ImageNet**, we use 1.1, 0.1, 0.001 as learning rate for SGDL, SGDM, and ADAM.

Table 1: Popular optimizers in the paper on CIFAR10, CIFAR100, ImageNet32 and Speech Commands Dataset. $\eta$ denotes momentum, $\gamma$ is the weight decay coefficient,$\beta$ is from Kingma & Ba (2014) and $\rho$ is from Zeiler (2012).

| Optimizer | Learning rate | Other Parameter Tuning |
|---|---|---|
| Adam | $\{0.0001, 0.001, 0.01\}$ | $\beta = (0.9, 0.999)$ |
| AdaDelta | $1.0$ | $\rho = 0.9$ |
| SGD | $\{0.001, 0.01, 0.05, 0.1\}$ | $\gamma = 0.005$ |
| SGDM | $\{0.001, 0.01, 0.05, 0.1\}$ | $\eta = 0.9, \gamma = 0.005$ |
| SGDL | $\geq 10 * \mathrm{lr}(\mathrm{SGDM})$ | $\gamma = 0.005$ |

### A.3.2 SPEECH COMMANDS DATASET

On Speech Commands Dataset, Similarly, the learning rate is 0.001 for Adam and 1.0 for AdaDelta as above. We choose 0.01 from $\{0.001, 0.01, 0.05, 0.1\}$ as the best for SGDM and 0.15 for SGDL 15 times larger than SGDM.

### A.3.3 DETAILS ON LSTM

For LSTMs, there are 1150 units in the hidden layer, an embedding of 400 and a batch size of 20. We select the best learning rate 1 from $\{0.001, 0.01, 0.1, 1\}$ for SGD(M) and the learning rate is same as above sections for Adam and Adadelta. For SGDL, we use 30 as an initial learning rate from the author's advise(Merity et al., 2017a) on $\{2, 3\}$-Layer LSTM for 150 epochs with the same annealing mechanism similarly as above. To train all models, we carry out gradient clipping with maximum norm 0.25. More specifically, for those dropout values, we use $(0.4, 0.3, 0.4, 0.1, 0.5)$ on word vector, output between LSTM layers, output of final LSTM layer, embedded dropout and DropConnect respectively.

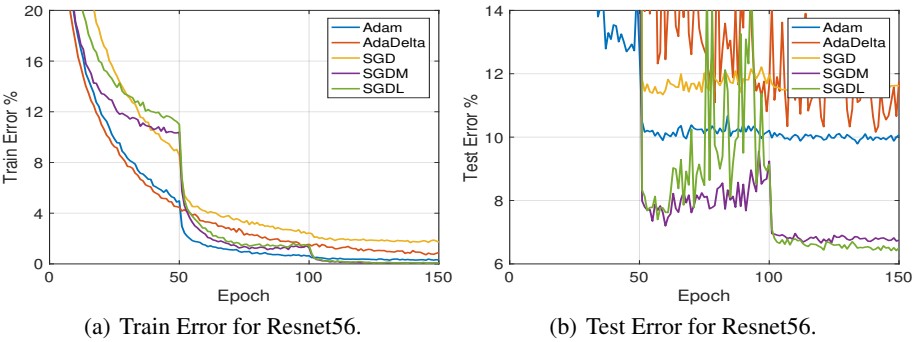

(a) Train Error for Resnet56.    (b) Test Error for Resnet56.

Figure 8: Resnet56 on CIFAR-10.

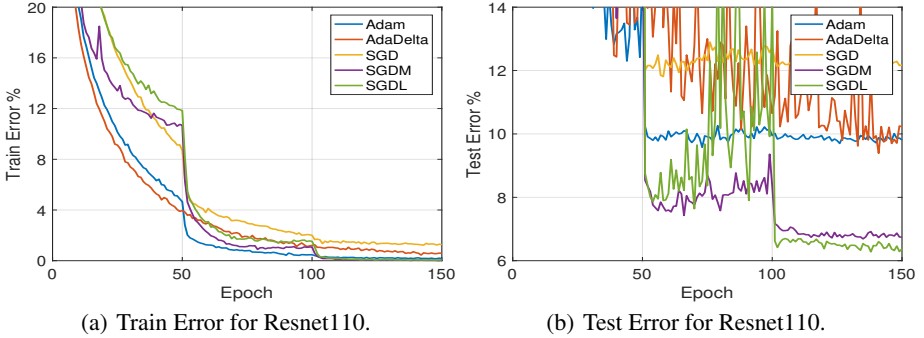

(a) Train Error for Resnet110.  (b) Test Error for Resnet110.

Figure 9: Resnet110 on CIFAR-10.

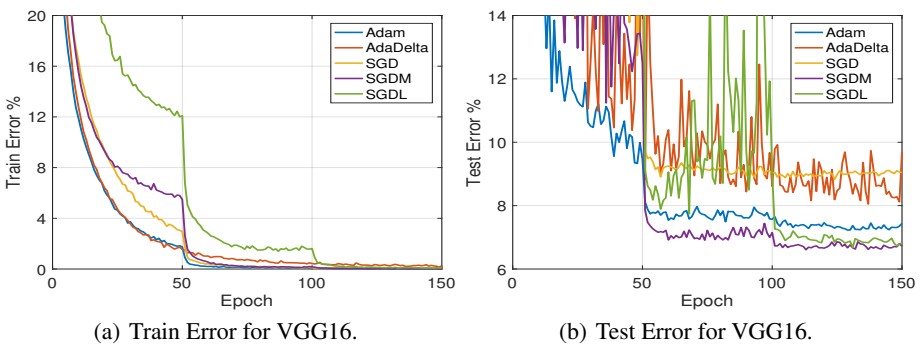

(a) Train Error for VGG16.  (b) Test Error for VGG16.

Figure 10: VGG16 on CIFAR-10.

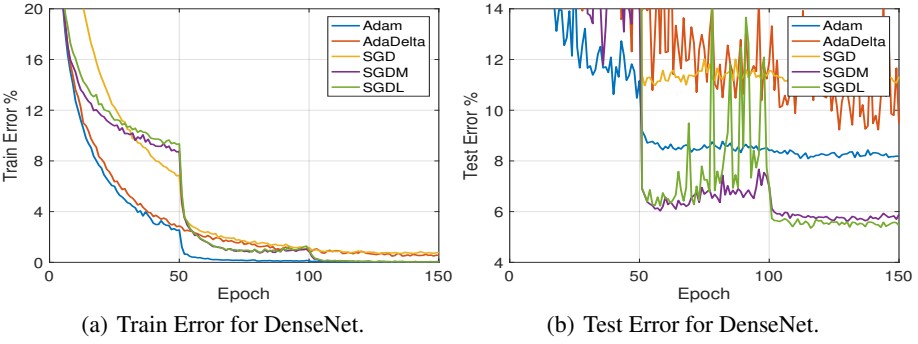

(a) Train Error for DenseNet.  (b) Test Error for DenseNet.

Figure 11: DenseNet on CIFAR10.

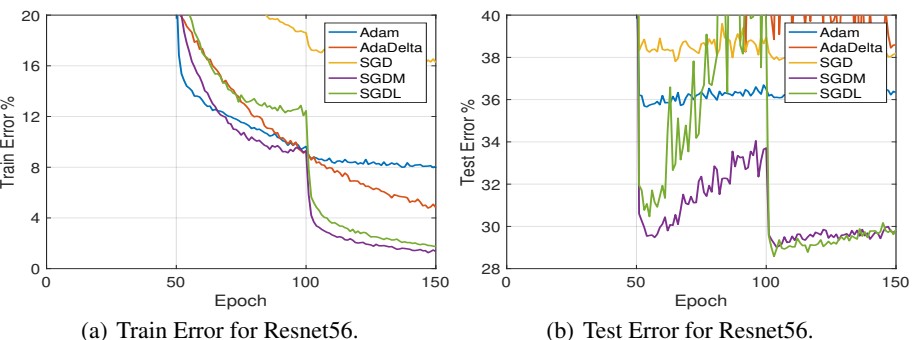

(a) Train Error for Resnet56.  (b) Test Error for Resnet56.

Figure 12: Resnet56 on CIFAR100.

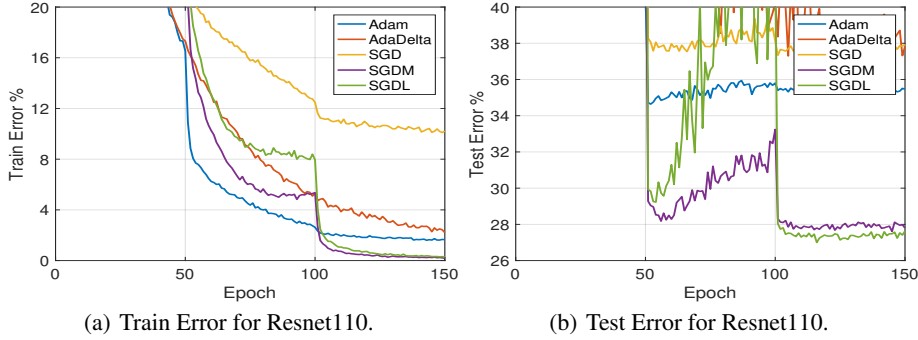

(a) Train Error for Resnet110.          (b) Test Error for Resnet110.

Figure 13: Resnet110 on CIFAR100.

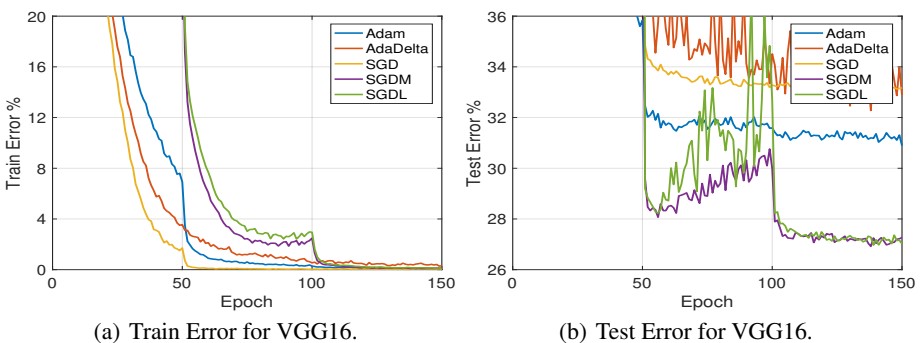

(a) Train Error for VGG16.          (b) Test Error for VGG16.

Figure 14: VGG16 on CIFAR100.

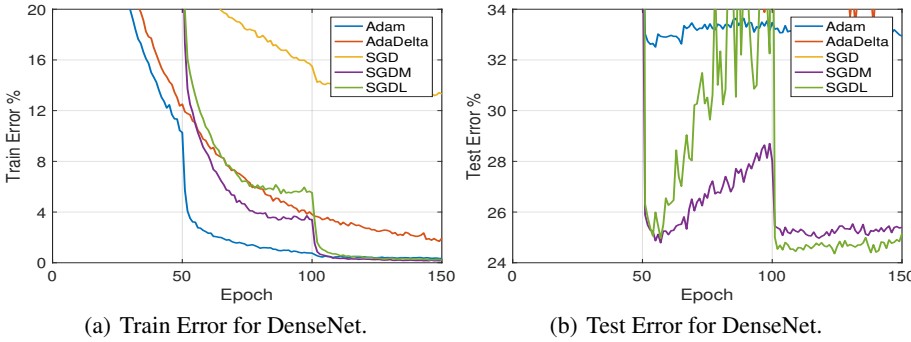

(a) Train Error for DenseNet.          (b) Test Error for DenseNet.

Figure 15: DenseNet on CIFAR100.

