# OpenReview forum: "Training By Vanilla SGD with Larger Learning Rates"
_ICLR.cc/2021/Conference — Reject_

### Official Review · AnonReviewer2 · 2020-10-15
**Weak analysis, strong and unrealisitc assumption, and unsupported claims**

**Rating:** 3
**Confidence:** 4

**Review:**

This paper investigates the SGD with constant step size (SGD-CS) on non-conex optimization problems. Theoretically, the paper shows the conditions under which a minimizer $x^*$ is a point of attraction in a local neighborhood under the algorithm SGD-CS with sufficiently small step-size. Furthermore, the paper experimentally shows that vanilla SGD-CS with relatively large step-size performs well, or even outperforms its momentum and/or adaptive counterparts, on several popular deep learning tasks.

[Comments]:

1: Claim and analysis are not consistent. The authors claim in the introduction that convergence of SGD-CS on non-convex functions is shown. However, the analysis only focuses on points of attraction and the stay within a local neighborhood. I would like to point out that the latter concepts are not equivalent to convergence. To show convergence of an algorithm, the missing piece of the paper is that, starting from the initialization point, the algorithm can be guaranteed to find such a local neighborhood around the point of attraction. Without this guarantee, the analysis of points of attraction is meaningless in the sense of algorithm convergence. Hence, I disagree that convergence of SGD-CS is theoretically shown in the paper.

2: Considering the necessary condition for points of attraction, Theorem 1, the assumption A.3 barely holds true in practical cases. As stated in Theorem 1, $x^*$ being a point of attraction implies interpolation property. For most real world tasks, interpolation can be only achieved when the model is over-parameterized, i.e., number of parameters is greater than the number of data samples. (For example, consider solving a system of linear equations). As pointed out by the work [Liu et al. 2020], most of the minimizers are not isolated, instead they form a low-dimensional manifold. In this case, none of the minimizers satisfies Assumption A.3, because the Hessian matrice H at the minimizers always have zero-eigenvalues (flat directions).

3: The main theoretical result, Theorem 2, relies on the fact that step-size is sufficiently small. However, one of the main claims of the paper is the convergence under large step size, as discussed in Section 5 and experimented in Section 6. I don’t see the connection between the small step-size theoretical result and the large step-size experiments. The theory seems not to explain the experiments.

4: The paper frequently talks about large learning rates. However, it is not clear to me what is the criteria to be large or small. Especially, in section 5, the paper provides a certain range of step size values (e.g., step size $t \in (0, 1/\lambda_{max}(H))$, within which the SGD-CS converges on a few simple examples. What are the reasons to claim these step-sizes are large?

[About clarity]:

1: It should be reader friendly to enlarge some of the figures.

2: Providing an intuition of the error function, defined in Eq.(4), should be helpful.

3: Notations can be improved.

[References]:

[Liu et al. 2020] Liu, Zhu, and Belkin. Toward a theory of optimization for over-parameterized systems of non-linear equations: the lessons of deep learning. arXiv:2003.00307.

---

> ### Author Response · Authors · 2020-11-21
> **Reply to AnonReviewer2**
>
> Thanks for your comments. All the points you mentioned are corrected in our new paper. You can refer to the blue highlighted text in the revised paper, and now we will explain the corrected points briefly in the following.
>
> 1.Thanks for your comments in the convergence analysis. SGD-CS cannot converge when starting from any initialization point in non-convex functions. In this work, our theoretical justify is to show the last-iterate convergence of SGD. The motivation is as follows: once the initial point is bad, and it takes a long distance to a neighborhood around a strong minimizer, then the step size is quite close to 0. We show that there is a scale of constant step size for convergence when it is in the neighborhood.
>
> 2.For assumptions: All assumptions are reasonable in some of the applications. The interpolation almost always holds in an over-parameterized DNN, which is useful for the computer vision and natural language model. For 8(c) Assumption(A.3), thanks to Review3’s reminder. We study a regularized problem in our paper by adding an L2 regularization in the objective function $f$ to make the Hessian $H(x^*)$ positive definite, and we also use L2-regularizer in all our experiments.
>
> 3.The connection between the theoretical result and experiments is by a case study in section5, which is a special example of non-convex optimization, and it shows that as long as the learning rate is bounded by a threshold, the larger learning rate, the smaller spectral radius (which means a faster convergence rate).
>
> 4.The scale of the learning rate for SGDL is discussed in the Appendix and now we add them in the Introduction section. Generally speaking, we say that a learning rate that is 10 times larger than that in SGD+momentum is a relatively large one. Decaying the learning rate happens in our experiments, but it does not hurt our definition of large, since the learning rate is still 10 times larger than that in SGD + momentum.  Many prior works have already shown better generalization for the first order optimization in comparison to the adaptive optimization(Adam), but to our understanding, the above experimental results focus on the SGD with momentum, and our paper shows SGD without any modification can still have a quite competitive performance.
>
> 5. We modified all points raised in the clarity part of your comment.

---

### Official Review · AnonReviewer1 · 2020-10-28
**interesting (albeit a bit disconnected) theoretical and empirical results**

**Rating:** 6
**Confidence:** 3

**Review:**

This paper presents a theoretical analysis of SGD with constant step size (SGD-CS) and presents conditions under which SGD-CS leads to parameter updates that converge to a local minima, including parameters of non-convex functions.  The authors then show, in context of some special functions, that the step size can be fairly large yet convergence is achieved.  This is followed by a number of empirical studies of SGD with large (but annealed) step-size (SGDL) on a variety of tasks.

I find the connection of SGD-CS with SGDL tenuous, and it is not clear that the theoretical analysis helps in selecting largest possible step size.  However, I do find the following contributions of the paper valuable:
a) Analysis of SGD-CS sheds lights on conditions under which the minimizers (local) of objective functions act as attractor (or strong attractor) of SGD updates.  This is worth sharing.
b) The empirical results with SGDL show a very consistent pattern of SGDL outperforming other optimization approaches (including ADAM, SGD with momentum, etc.).  While this is not really SGD-CS, I find the consistent behavior of SGDL worth noting and sharing.

Some clarifications and minor typographical errors:
* SGC in last sentence of Section 3.1 is not defined.
* Section 3.2 assumptions A.1 and A.2 should refer to Eq. (1b) and not (1a) I think
* In Figure 2 which optimization approach is used to derive the curves?
* In Fig. 6, the two curves in (a) and two in (b) … are they train and test perplexity results?
* Last sentence of conclusions is unclear, please restate.

---

> ### Author Response · Authors · 2020-11-21
> **Reply to AnonReviewer1**
>
> Thanks for your comments. All the points you mentioned are corrected in our new paper. You can refer to the blue highlighted text in the revised paper, and now we will explain the corrected points briefly in the following.
> For clarify:
> 1.	We have added the SGC in our related work now.
> 2.	In Figure 2, we use vanilla SGD with different learning rates. Yes, there are train and test perplexity results.
> 3.	Yes, they are train and test perplexity.
> 4.	We rewrite the last sentence.

---

### Official Review · AnonReviewer4 · 2020-10-28
**Reviews for The simpler the better: vanilla sgd revisited**

**Rating:** 5
**Confidence:** 4

**Review:**

This paper studies the smooth finite-sum problem under suitable conditions in the non-convex case. They show the necessary condition for the minimizer $x^*$ being a point of attraction, and Theorem 1 provides a sufficient condition for the strong minimizer $x^*$ to be a point of strong attraction with high probability. Based on the results, they introduce a modified SGD algorithm with a large initial learning rate (SGDL), and provide extensive experiments on various popular tasks and models in computer vision, audio recognition and natural language processing.
pros: 1, They give a sufficient condition for the strong minimizer $x^*$ to be a point of strong attraction with high probability.
2, Extensive experiments are presented to show the effectiveness of SGDL.

cons: 1, In Theorem 2, it is better to show how to choose the $\epsilon$ explicitly and what linear convergence rate can be achieved, i.e., how small the parameter $\rho$ can be.
2, Even though this paper considers the non-convex case, the assumptions seems very restrictive. Assumption A.3 means that in a neighborhood of $x^*$, the objective function is essentially strongly convex. Furthermore, all $\nabla f_i(x^*)$ need to be zero.

minor comment: In the first inequality of the proof of Remark 2, why the bound is not zero? Since under the condition $\tau=\infty$, $||X_k-x^*||$ should be no larger than $\epsilon$.

---------------------After the rebuttal------
The authors partially addressed my concerns. I remain the current score.

---

> ### Author Response · Authors · 2020-11-21
> **Reply to AnonReviewer4**
>
> Thanks for your comments. All the points you mentioned are corrected in our new paper. You can refer to the blue highlighted text in the revised paper, and now we will explain the corrected points briefly in the following.
>
> 1.	How to choose $\epsilon$ was discussed in the Appendix previously, and now we move it into the main text (Theorem 2).
> 2.	For assumptions: All assumptions are reasonable in some of the applications. The interpolation almost always holds in an over-parameterized DNN, which is useful for the computer vision and natural language model. For 8(c) Assumption(A.3), thanks to Review3’s reminder. We study a regularized problem in our paper by adding an L2 regularization in the objective function $f$ to make the Hessian $H(x^*)$ positive definite, and we also use L2-regularizer in all our experiments.
> 3.	Thanks for your minor comment. Now we corrected the proof of Remark 2 and the probability is at least $1-\delta$.

---

### Official Review · AnonReviewer3 · 2020-10-28
**The paper lacks of innovation and theoretical justification for their claiming points.**

**Rating:** 4
**Confidence:** 5

**Review:**

Main idea: As a classical and effective optimizer, vanilla SGD can always compete with or even outperform its momentum or adaptive variations when training over-parameterized DNNs. The paper aims to theoretically justify this claim and empirically compare the performance across multiple tasks.

1.	What is exactly the overall advantage or difference between the SGDL and the vanilla SGD? Particularly, in how to choose the stepsize alpha?
2.	There is no theoretical comparisons between SGD and its momentum and adaptive variations. Because the paper claims to theoretically justify the claim that “SGD is better”, can authors point out how they justify it theoretically?
3.	Remark 1 that follows theorem 2 gives particular conditions to make Eq.(7) hold. Can authors explain more on how to derive these 3 conditions 8(a)-(8(c)?  Does the proof assume that the initial point is close enough to the optimal solution? How can SGDL globally converge (converge from any starting point)?
4.	In experiments, what is SGDM?
5.	In experiments, the paper uses decaying learning rate, so a large initial stepsize can quickly decay into a small number, so how does this become an advantage?  SGD has better generalization which has been observed in many prior works.
6.	The paper has some typos, and the meaning of some sentences is puzzling.  For instance, (1) there are multiple uses of N in definition 2; (2) the index used in the paper is not consistent, i=1:N and i=1,…,N are both used in the manuscript and other format has been used in the proofs; (3) Eq.8(b) and the condition of theorem 2 are not consistent.

#####################
update:  I have read authors' response to my comments and also read other reviewers' comments and discussions.  The main concern of my comments is still not clear. I will keep my rating unchanged.

---

> ### Author Response · Authors · 2020-11-21
> **Reply to AnonReviewer3**
>
> Thanks for your comments. Almost all the points you mentioned are corrected in our new paper. You can refer to the blue highlighted text in the revised paper, and now we will explain the corrected points briefly in the following.
>
> 1.SGDL means the vanilla SGD with a relatively large learning rate. The scale of the learning rate for SGDL is discussed in the Appendix and now we add them in the Introduction section. Generally speaking, we say that a learning rate that is 10 times larger than that in SGD+momentum is a relatively large one.
>
> 2.For proof:  SGD-CS cannot converge when starting from any initialization point in non-convex functions. In this work, our theoretical justification is to show the local ( last-iterate) convergence of SGD. The motivation is as follows:  once the initial point is bad, and it takes a long distance to a neighborhood around a strong minimizer, then the step size is quite close to 0. We show that there is a scale of constant step size for convergence when it is in the neighborhood.
>
> 3.For assumptions: All assumptions are reasonable in some of the applications. The interpolation almost always holds in an over-parameterized DNN, which is useful for the computer vision and natural language model. For 8(c) Assumption(A.3), thanks to Review3’s reminder. We study a regularized problem in our paper by adding an L2 regularization in the objective function $f$ to make the Hessian $H(x^*)$ positive definite, and we also use L2-regularizer in all our experiments.
>
> 4.SGDM is SGD with momentum.
>
> 5.Decaying the learning rate happens in our experiments, but it does not hurt our definition of large, since the learning rate is still 10 times larger than that in SGD + momentum.  Many prior works have already shown better generalization for the first order optimization in comparison to the adaptive optimization(Adam), but to our understanding, the above experimental results focus on the SGD with momentum, and our paper shows SGD without any modification can still have a quite competitive performance.
>
> 6.Sincerely thanks for pointing out some typos. We have already modified it in our new version.
>
> 7.unsolved one: For SGD and SGD+momentum, it is difficult to justify SGD is better than SGD + momentum theoretically, which is beyond the scope of this work. However, the extensive experimental results in this paper have already shown that vanilla SGD still can be used and its performance is quite comparable to its modern variants in many cases.

---

### Decision · Program_Chairs · 2021-01-07
**Final Decision**

**Decision:**

Reject

**Comment:**

The paper primary theoretical contribution claim is to establish the constant size SGD converges linear to the optimal solution in non-convex settings. This is shown in the interpolation regime for over-parametrized situations when starting from points nearby to the optimum. The paper's empirical claim is to use relatively larger learning rates for SGD in common deep learning settings and claim that they can do well.

My recommendation is based on the overall low scores provided by the reviewers - which did not change post rebuttal. The concerns raised by the reviewers amounting to my decision recommendation is summarized below -

Overall the reviewers found the connection between the theoretical results and the overall claims of the paper unconnected. The reviewers found the theoretical contribution of the local convergence weak - particularly in the context of an analysis of constant learning rates and taking into account existing work on the convex case for such results. Furthermore, the experimental contribution of the paper is incremental as the proposed algorithm is standard with just a larger than typical initial learning rates. This factor is usually searched over during Hyper Parameter sweeps in all the large scale learning setups. In this context, SGDL performing favorably, is an interesting observation but not enough of a contribution. Further the reviewers objected to the fact that SGDL does not connect with the theory presented as SGDL in experiments still uses learning rate schedules.